# Is Hearing Loss a Risk Factor for Idiopathic Parkinson’s Disease? An English Longitudinal Study of Ageing Analysis

**DOI:** 10.3390/brainsci13081196

**Published:** 2023-08-12

**Authors:** Megan Rose Readman, Fang Wan, Ian Fairman, Sally A. Linkenauger, Trevor J. Crawford, Christopher J. Plack

**Affiliations:** 1Department of Psychology, Lancaster University, Lancaster LA1 4YW, UK; 2Department of Primary Care and Mental Health, The University of Liverpool, Liverpool L69 3BX, UK; 3NIHR ARC NWC, Liverpool L7 8XP, UK; 4Department of Mathematics and Statistics, Lancaster University, Lancaster LA1 4YW, UK; 5Public Advisor, Associated with Lancaster University Psychology Department, Lancaster LA1 4YF, UK; 6Manchester Centre for Audiology and Deafness, University of Manchester, Manchester M13 9PL, UK

**Keywords:** Parkinson’s disease, hearing loss, risk factor, sensory processes

## Abstract

Observations that hearing loss is a substantial risk factor for dementia may be accounted for by a common pathology. Mitochondrial oxidative stress and alterations in α-synuclein pathology may be common pathology candidates. Crucially, these candidate pathologies are implicated in Parkinson’s disease (PD). Consequently, hearing loss may be a risk factor for PD. Subsequently, this prospective cohort study of the English Longitudinal Study of Ageing examines whether hearing loss is a risk factor for PD longitudinally. Participants reporting self-reported hearing capabilities and no PD diagnosis prior to entry (*n* = 14,340) were used. A joint longitudinal and survival model showed that during a median follow up of 10 years (SD = 4.67 years) increased PD risk (*p* < 0.001), but not self-reported hearing capability (*p* = 0.402). Additionally, an exploratory binary logistic regression modelling the influence of hearing loss identified using a screening test (*n* = 4812) on incident PD indicated that neither moderate (*p* = 0.794), nor moderately severe/severe hearing loss (*p* = 0.5210), increased PD risk, compared with normal hearing. Whilst discrepancies with prior literature may suggest a neurological link between hearing loss and PD, further large-scale analyses using clinically derived hearing loss are needed.

## 1. Introduction

Hearing loss is oftentimes considered an inevitable sequela of ageing. However, an accumulation of evidence has revealed that hearing loss is a substantial, potentially modifiable, risk factor for incident all-cause dementia [1]. Specifically, Lin et al. [2] demonstrated that mild hearing loss almost doubles dementia risk, moderate hearing loss triples dementia risk, and severe hearing loss increases dementia risk by almost five times, over ~12 years of follow up.

The underlying mechanism relating hearing loss and dementia remains unclear. The common cause hypothesis suggests that a common pathology affects both the cochlea and ascending auditory pathway (causing hearing loss) and the cortex (causing dementia) [3,4,5]. Mitochondrial oxidative damage and alterations in the production, and aggregation, of α-synuclein are two potential common pathology candidates. Following the rationale of the common cause hypothesis, it may be that hearing loss is a significant risk factor for alternative clinical conditions in which mitochondrial oxidative damage and alterations in the production, and aggregation, of α-synuclein are implicated, such as the progressive neurodegenerative disorder Parkinson’s Disease (PD).

Oxidative stress, a biological state, occurs when the production and accumulation of reactive oxygen species (ROS) within cells outweighs detoxification of these molecules [6]. This state initiates damage to cellular macromolecules, including mitochondrial membranes, respiratory chain proteins, and nuclear and mitochondrial DNA [7], which can eventually trigger upstream cell apoptosis [8]. Evidence indicates that mitochondrial oxidative stress is implicated in dementia pathogenesis [9]. For example, increased levels of DNA strand breaks [10,11], and brain lesions [12], characteristic of oxidative stress, have been observed in Alzheimer’s disease. Similarly, mitochondrial oxidative damage in the cochlea is also implicated in the pathogenesis of acquired hearing loss [13,14]. Specifically, mitochondrial DNA alterations, indicative of oxidative stress, in the cochlear tissue are more frequent in patients with age-related hearing loss compared to those with normal hearing [15]. Furthermore, genetic variations in certain antioxidant defence genes have been associated with increased risk of age-related hearing loss [16,17,18].

PD is pathologically hallmarked by accelerated dopaminergic neuronal cell death, particularly within the substantia nigra of the basal ganglia [19]. Whilst the mechanism underlying this is not yet elucidated, several patomechanisms have been hypothesised. One of these hypotheses focuses on the role of chronic neuroinflammation, microglial activation and the subsequent oxidative stress [20]. Indeed, evidence suggests that regular non-steroidal anti-inflammatory drug (NSAID) users are at reduced risk (~36–55 percent reduced) of PD [20]. Moreover, significantly elevated inflammation levels are observed in the blood, identified through neutrophil-to-lymphocyte ratio and platelet-to-lymphocyte ratios, and the cerebrospinal fluid of PD patients relative to controls [21]. Importantly, it is postulated that activated microglia produce large amounts of reactive oxygen species (superoxide radicals) thus implicating oxidative damage in PD [22]. Supporting this assertion, consumption of MPTP (1-methyl-4-phenyl-1,2,3,6-tetrahydropyridine), which increases ROS production and leads to a sustained inflammatory reaction, can induce Parkinsonism [23]. Comparably, elevated circulating oxidative stress markers (e.g., ferritin, 8-OhdG and nitrite) and decreased antioxidant levels (e.g., catalase, uric acid, glutathione) have been observed in PD [24,25].

Another potential common pathological candidate, for the relation between hearing loss and dementia, is alterations in the production, and aggregation, of α-synuclein. Natively, α-synuclein is unfolded. However, in response to specific environmental factors, such as toxins and inflammation, α-synuclein folds into monomers, polymers, and oligomers [26]. Notably, folded α-synuclein are major components of Lewy bodies [27].

Importantly, α-synuclein Lewy body pathology has been detected in over 50 percent of sporadic [28,29] and ~60 percent of familial [30], Alzheimer’s disease cases at autopsy. Similarly, elevated levels of soluble α-synuclein are observed in the cerebral spinal fluid of patients with Alzheimer’s disease [31] and mild cognitive impairment [32,33] relative to controls. Regarding hearing loss, Park et al. [34] observed weaker efferent nerve and cochlear homogenate α-synuclein expression in early-onset hearing loss, compared to late-onset hearing loss mouse models. Furthermore, Akil et al. [35] found significantly elevated auditory brain stem thresholds in α-synuclein overexpression transgenic mice compared to wild-type mice, implicating α-synuclein overexpression in hearing loss.

Critically, in addition to accelerated dopaminergic neuronal degeneration, PD is also neuropathologically hallmarked by intracellular Lewy body inclusions and Lewy neurites [19]. Moreover, mutations (e.g., PARK1 and PARK4) in the α-synuclein gene (SNCA) are accountable for autosomal dominant PD [36], and variations in the SNCA gene appear to be one of the strongest risk factors in the development of sporadic PD [37]. Subsequently, α-synuclein may contribute to the neuropathology of PD [38].

Thus far, mitochondrial oxidative damage and alterations in α-synuclein aggregation have been presented as dissociable pathologies of hearing loss and dementia. However, these pathologies may form a positive feedback loop in which either event may trigger a self-perpetuating process [39,40,41]. Given that mitochondrial oxidative stress and alterations in α-synuclein production and aggregation are central to PD pathogenesis [42], following the rationale of the common cause hypothesis [3], we may hypothesise that hearing loss will be intricately related to PD.

Consideration of the relation between essential tremor and hearing loss may further support the assumption that hearing loss will be intricately related to PD. Evidence suggests that patients with essential tremor, which can be defined as “bilateral upper extremity action tremor” [43], are at an elevated risk of PD [44]. Specifically, some analyses report that patients with essential tremor are up to four times more likely to develop PD than people without essential tremor [45]. Importantly, hearing loss appears to also be present in essential tremor patients [46]. Some studies have shown that essential tremor patients have significantly elevated pure tone audiometry thresholds at high but not low frequencies [47], and others have observed significantly elevated pure tone audiometry thresholds at low but not high frequencies [48]. Given that essential tremor is a risk factor for PD, and patients with essential tremor are likely to experience hearing loss, we may predict that hearing loss is likely to proceed PD.

Indeed, a meta-analytical review concluded that sensorineural hearing loss and cochlear impairment are more severe in PD patients compared to age-matched controls [49]. Moreover, asymptomatic hearing impairment (when indicators of hearing loss are observed without self-reported hearing difficulties) is significantly higher in PD patients than age-matched controls [50,51]. Consequently, altered audiological function may be related to PD [52]. However, whether hearing loss results from complex sensory processing impairments during PD, or whether hearing loss antedates PD diagnosis, and is therefore a risk factor for PD, remains unclear.

Recent prospective studies indicate that hearing loss may antedate, and be a risk factor for PD. For example, Schrag et al. [53] found that hearing loss, up to 5 years pre-diagnosis, was more prevalent amongst those who later developed PD than controls. Moreover, Lai et al. [54] found that, over a five-year follow-up, hearing loss increases the risk of incident PD by 1.5-fold. Similarly, Simonet et al. [55] observed that hearing loss increases PD risk by 1.6-fold over <2 years follow-up. These studies implicate hearing loss as a risk factor for subsequent PD diagnosis; however, further research is required to substantiate these claims within alternative populations (i.e., using different databases/cohorts). Moreover, these studies use clinical hearing loss diagnosis (i.e., a documented medical diagnosis) as a dependent measure. As such, whether hearing loss at different neurophysiological levels, using different measures of hearing loss, antedates PD diagnosis, and so may be a risk factor for PD, remains unclear. This study aims, therefore, to determine whether hearing loss is a risk factor for later PD diagnosis using data from the English longitudinal study of ageing (ELSA) [56].

## 2. Materials and Methods

This study was pre-registered on the Open Science Framework (OSF; https://doi.org/10.17605/OSF.IO/MJGQ6, accessed on 1 February 2023), and the data analysed in this study are openly available in the UK Data Service at https://doi.org/10.5255/UKDA-SN-5050-25, accessed on 18 November 2022. The OSF page for this project includes details regarding the variables analysed, planned statistical analyses, and the data analysis code book. We summarise the implementation below. This study deviated from the pre-registration only in the number of covariates embedded within both the primary and exploratory analyses and the exclusion criteria for missing data. Please refer to Appendix B for full justification for these deviations.

### 2.1. Study Population

#### 2.1.1. Primary Analysis

The ELSA is an ongoing longitudinal prospective cohort study of adults living in private households in England [56]. Data collection follows a longitudinal design with repeated measures of core variables occurring biannually over numerous ‘waves’. The present study utilised the full dataset, including core participants and younger and older responding relatives, from waves 1–9 (see Appendix C for further sampling details). Participants were excluded if they:1.Reported a diagnosis of PD at wave 1;2.Appeared only in one wave;3.Were missing data for self-reported hearing capabilities, age, or PD diagnosis variables.

The final analytical sample was *n* =14,340 (6466 males; 7874 females).

#### 2.1.2. Post-Hoc Exploratory Analysis

We used HearCheck^TM^ Screener data from wave 7 and incident PD data from wave 9 of ELSA. Participants reporting a PD diagnosis at wave 7 who did not complete the HearCheck^TM^ Screener test, or who did not have age data available, were excluded. The final analytical sample was *n* = 4812 (2065 males, 2747 females).

### 2.2. Outcome Measure

#### 2.2.1. Primary Analysis

Participants disclosed whether they had a diagnosis of PD at all waves. PD diagnosis was confirmed at subsequent waves.

#### 2.2.2. Post-Hoc Exploratory Analysis

Participant reported PD in wave 9.

### 2.3. Exposure Measure

#### 2.3.1. Primary Analysis

Participants self-reported hearing capabilities at each wave by indicating on a five-point Likert scale “Is your hearing [using a hearing aid as usual] (1) excellent, (2) very good, (3) good, (4) fair, or (5) poor”.

#### 2.3.2. Post-Hoc Exploratory Analysis

The HearCheck^TM^ screener generates a series of six tones: three mid-frequency tones of decreasing volume at 1 kHz (55, 34, and 20 dB) and three high-frequency tones of decreasing volume at 3 kHz (at 75, 55, and 35 dB). Participants indicate when they hear the tone by raising their finger. Audibility testing occurs separately for each ear. Participants are required to remove any hearing aid(s) prior to completion of the screening test, thus the obtained results reflect hearing capabilities in the absence of corrective hearing devices. Evidence indicates that hearing loss, defined as >35 dB at 3 kHz in the better-hearing ear, is a robust and justifiable cut off criterion at which intervention is beneficial [57,58]. Subsequently, hearing loss was defined according to this criterion. The resulting hearing loss variable was further subdivided into two mutually exclusive categories: (1) ‘moderate hearing loss’: >35 dB to 54 dB (tone not heard at 34 dB HL but heard at 55 dB and at 75 dB) and (2) ‘moderately severe or severe loss’: >55 dB HL (tone not heard at 35 dB HL and at 55 dB HL, but the tone may, or may not, have been heard at 75 dB HL).

### 2.4. Covariates

Age is the biggest risk factor for developing PD [59] and hearing loss [60]. Hence, we controlled for age, as a continuous variable, in all analyses. Age at entry into ELSA is used as a baseline covariate, while time spent in ELSA (derived from wave number) is used as the timescale for time to diagnosis of PD and the longitudinal HL outcomes.

### 2.5. Data Analysis

#### 2.5.1. Primary Analysis

Joint longitudinal and survival models consider the potential dependency between the longitudinal and survival data [61], assessing the impact a longitudinal covariate, measured with error, has on the time to an event of interest [62]. Hence, a joint longitudinal and survival model was fitted to assess the relation between self-reported hearing loss and subsequent PD, with age as a covariate.

Model structure. There are two components to a joint model: a longitudinal component and a time-to-event survival component. The longitudinal and survival components of the model were joined through a trajectory function. (see Appendix D for full joint model mathematical formulae).

Longitudinal component. The longitudinal component consisted of a continuation ratio model for the ordinal hearing loss variable, which assumes a linear model with random effects for the log odds of being in the kth category, conditional on being in the kth category or higher.

Survival component. The survival component, and time to PD diagnosis, of the joint model consisted of a Weibull model.

#### 2.5.2. Post-Hoc Exploratory Analysis

Self-reported hearing capabilities are oftentimes influenced by non-auditory factors, including demographic and socioeconomic characteristics (e.g., biological sex and educational attainment [63] and medical history [64]). Indeed, almost one-third of participants with screening-test-identified hearing loss in ELSA wave 7 went undetected by self-report measures [65], indicating a low concordance between self-reported and screening test hearing capabilities. The wave 7 ELSA screening test hearing loss measure is obtained from the Siemens HearCheck^TM^ Screener. The HearCheck^TM^ Screener yields a sensitivity of 0.89 and specificity of 0.86 [66], thus may constitute a more reliable measure of hearing loss. Hence, we conducted a further exploratory analysis to determine whether screening-test-derived hearing loss is a risk factor for subsequent PD diagnosis. The HearCheck^TM^ screening test is not conducted at every wave, rather it was only conducted at wave 7. As such, only one HearCheckTM Screener data point is currently available per participant, thus not permitting a longitudinal analysis. Therefore, to determine whether screening-test-derived hearing loss is a risk factor for subsequent PD diagnosis, a case-control binary logistic regression was conducted.

## 3. Results

### 3.1. Primary Analysis

#### 3.1.1. Sample Demographics

During a median follow up of 10 years (SD = 4.67 years), 151 cases of incident PD were reported in the analysed sample (*n* = 14,340), with 89 of these being male.

#### 3.1.2. Joint Longitudinal and Survival Model

A notable proportion of participants’ self-reported hearing capabilities responses remained stable between two wave points (e.g., wave 7 to wave 8), with the average proportion (across all nine waves) of participants reporting the same hearing capabilities at two consecutive waves being 0.46 for excellent, 0.43 for very good, 0.49 for good, 0.46 for fair and 0.42 for poor. A small number of participants reported their hearing capabilities at a given wave to be better than the prior wave. This proportion marginally increased as self-reported hearing capabilities decreased. That is, a marginally larger proportion of participants reported their hearing capabilities as being better at a subsequent wave if they reported their hearing capabilities to be fair or poor at the prior wave, than participants who reported their hearing to be good, very good, or excellent at the prior wave. Across all waves, a substantial proportion of participants reported their hearing capabilities at a given wave to be worse than the prior wave. Of these participants, most reported their hearing capabilities to be one rating scale point lower than the prior wave (i.e., from excellent at wave 1 to good at wave 2). However, some participants did report a reduction of two or three rating scale points. A full breakdown of the stability of self-reported hearing capabilities can be found in the Appendix A.

Longitudinal outcome. Both age at entry into ELSA (*p* < 0.001) and length of time since entering the study (time in study, *p* < 0.001) were significantly associated with self-reported hearing loss. Specifically, for each unit increase in either age or time since entering the study, the odds of being in a better self-reported hearing category decreases by 5.7 percent (95 CI: 5.4–6.0).

Survival outcome. Concerning incident PD diagnosis, as age increases, the likelihood of PD diagnosis increases (*p* < 0.001). However, self-reported hearing capability was not related significantly to subsequent PD diagnosis (*p* = 0.402) (see Table 1).

### 3.2. Post-Hoc Exploratory Analysis

#### 3.2.1. Sample Demographics

Over the ~4 year follow up (the duration between completion of wave 7 and 9), 24 (13 male, 11 female) cases of incident PD were reported in the analysed sample (*n* = 4812), with the mean age of people with PD being 74.67 (6.81).

#### 3.2.2. Binary Regression

In accordance with the HearCheck^TM^ screener, 3575 participants were categorised as having ‘normal hearing’, 850 participants were categorised as having ‘moderate hearing loss’, and 387 participants were categorised as having ‘moderately severe/severe hearing loss’. Generally, participants with greater hearing loss were more likely to be older and male. See Table 2 for the distribution of demographic characteristics of participants categorised by HearCheck^TM^ screener derived HL.

The assumption of linearity for the age variable was assumed by observing conditional density plots and using the Box Tidwell test (z = 0.316, *p* = 0.752). After controlling for age, compared with normal hearing, neither moderate hearing loss [OR = 0.87 (95 CI: 0.27–2.41), *p* = 0.794], nor moderately severe/severe hearing loss [OR = 1.45 (95 CI: 0.43–4.31), *p* = 0.521], increased risk of incident PD over a four-year follow-up.

## 4. Discussion

The present study examined whether hearing loss is a risk factor for incident PD. Regarding the incidence of PD, we observed that self-reported hearing capability was not a significant risk factor for incident PD over ~10 years of follow up. Similarly, HearCheck^TM^ screener-derived hearing loss was not a significant risk factor for incident PD over ~4 years. These observations may indicate that interventions for hearing loss (i.e., hearing aids or cochlear implants) may not affect PD progression. However, due to fundamental limitations in the dataset analyses (e.g., lack of data regarding hearing aid use and PD subcategories further research is needed to confirm this hypothesis. In terms of the longitudinal outcome (self-reported hearing capabilities) we observed that as age and length of time in the study increased, self-reported hearing capabilities decreased. Age-related hearing loss is characterised by degenerative pathology in the cochlear hair cells, supporting cells, and auditory nerve endings [67], and age is the largest risk factor for hearing loss [60]. Thus, this observation is to be expected. However, as it is not possible to manipulate increases in chronological age, it is implausible for this factor to be a clinical intervention target.

Although the present study did not observe a relation between hearing loss and incident PD, some previous literature suggests that hearing loss increases the future risk of incident PD by 1.14–1.6 fold [53,54,55]. Critically, this prior literature has relied upon clinically diagnosed hearing loss as the exposure measure. Within the localities in which these studies recruited participants (Taiwanese, English, and German populations), the clinical diagnosis of hearing loss relies upon pure tone audiometry assessments [68,69]. Previous literature has shown that the concordance between pure tone audiometry and self-reported hearing capabilities is low [65]. Hence, this discrepancy in hearing loss measures may explain the lack of concordance in the results.

Exploratory analyses also found that screening-test-derived hearing loss was not a significant risk factor for incident PD. Yet, whilst the HearCheck^TM^ screener constitutes a substantially more objective hearing loss measure than self-report questionnaires, the accuracy and sensitivity of the HearCheckTM are, at the lowest, 83 percent [66]. Consequently, discrepancies with prior research could result from use of a screening test measure with lower accuracy and sensitivity. Crucially, these exploratory findings should be treated with caution due to the small number of incident PD cases reported (*n* = 24). Additional larger-scale cohort analyses that draw on a more reliable diagnosis of hearing loss (e.g., pure tone audiometry) are required to further elucidate whether hearing loss is a risk factor for incident PD.

The discrepancies between the current findings and those of previous research could speak to the mechanism underlying hearing loss and antedating PD motor manifestations. Pure tone audiometry assessment is thought to depend primarily upon the health of the outer hair cells, and additionally cochlear transduction by the inner hair cells [70]. Although pure tone audiometry can be influenced by psychophysiological factors (e.g., patient attention) and demographic factors (e.g., biological sex) [71], it is largely assumed that pure tone audiometry outcomes primarily derive from peripheral sensory processes. In contrast, self-reported hearing capabilities are heavily influenced by non-auditory factors including demographic (e.g., biological sex, age), socioeconomic (e.g., educational level, occupation), and lifestyle factors (e.g., tobacco and alcohol consumption and physical inactivity [65]), as well as social stigma [72]. Hence, self-reported hearing loss may reflect higher-order subjective processes. Given that audiometrically derived [53,54,55], but not self-reported, hearing capabilities appear to be a risk factor for incident PD, this may suggest that hearing loss and PD are potentially related at a neurological level. This is consistent, potentially, with the speculations regarding the molecular basis of the common cause hypothesis.

Importantly, both the present study and previous research [53,54,55] have exclusively focused on whether hearing loss is a substantial risk factor for incident PD. Parkinsonism is an umbrella term used to describe a group of conditions clinically characterised by motor impairments including tremors, slowed movement, muscle rigidity, gait alterations, and postural instability [73]. Idiopathic PD is the most common Parkinsonism disorder, accounting for 85 percent of neurodegenerative cases. The remaining 15 percent of cases are attributed to atypical Parkinsonian disorders including Progressive Supranuclear Palsy (PSP), Multiple Systems Atrophy (MSA), Dementia with Lewy Bodies (DLB), and Corticobasal Degeneration (CBD). On a neuropathological level, atypical Parkinsonism disorders share several similarities with PD. For example, MSA and DLB, akin to PD, are α-synucleinopathies [74]. Moreover, neuroinflammation and oxidative stress have been implicated as patomechanisms in MSA [75,76], PSP [77,78] and CBD [79,80] (note that the evidence implicating oxidative stress in CBD is less extensive). However, whilst some studies have found DLB to be characterised by a lack of neuroinflammation [81], alternative studies have observed oxidative damage in DLB patients [82].

The present study observed that self-reported and HearCheck^TM^ screener-derived hearing loss are not risk factors for incident PD. However, if clinically diagnosed hearing loss is a risk factor for incident PD [53,54,55], it may be that, due to the neuropathological similarities between PD, PSP, MSA, DLB, and CBD, clinically diagnosed hearing loss will also be a risk factor for other Parkinsonism disorders. Alternatively, with respect to the common cause hypothesis, it may perhaps be that hearing loss is a risk factor for some, but not all, Parkinsonism disorders. Specifically, if hearing loss is related to neurodegenerative disorders through a neuroinflammation and oxidative stress common cause, we may anticipate that hearing loss will also be a risk factor for PSP, MSA, DLB, and CBD. However, if the link is driven by alterations in the production, and aggregation, of α-synuclein, then we would perhaps expect hearing loss to be a risk factor for MSA and DLB, but not PSP or CBD (as both are tauopathies [74]).

Although hearing loss may play a small role in the clinical manifestation of Parkinsonism disorders, observations of a differential influence of hearing loss on symptom onset across Parkinsonism disorders may be useful for clinicians to consider when differentiating such symptomatically similar conditions. Furthermore, ascertaining the relation between hearing loss and multiple Parkinsonism disorders may provide invaluable insights for the differential clinical management of Parkinsonism disorders. Therefore, the predictions made here should be investigated formally.

Within this study, PD was indicated by self-disclosure of a clinical diagnosis. In the UK, in accordance with UK Parkinson’s Disease Society Brain Bank guidelines, a probable diagnosis of PD is provided if any given individual presents with bradykinesia (slowed movement) and at least one of the following: muscular rigidity, rest tremor, or postural instability [83]. Whilst diagnostic criteria emphasise motor manifestations, cognitive disturbances are increasingly recognised as commonplace in PD. Specifically, 30 percent of people with PD will experience dementia [84]. At a pathological level, Parkinson’s dementia is oftentimes coupled with Alzheimer’s disease-related pathologies [85]. Given the evidence that hearing loss is a risk factor for incident all-cause dementia [1], it may be that hearing loss forms a risk factor for the occurrence and severity of cognitive impairment in PD, rather than overall diagnosis.

Clinical heterogeneity is well recognised in PD. Such heterogeneity presents a major barrier in terms of understanding disease mechanisms, developing treatments, and clinical management. Consequently, a set of clinical criteria for PD subtype categorization has been developed. The most frequently referred to subtyping system categorises patients into three subtypes: (1) mild-motor predominant, (2) diffuse malignant, and (3) intermediate, with categorization being based on motor impairment, cognitive function, rapid eye movement behaviour disorder, and autonomic symptoms [86]. Given that specific motor and non-motor symptoms are more strongly associated with some subcategories than others, it may be that hearing loss is a stronger risk factor for PD onset for certain subtypes over others. As previously described, PD diagnosis is a binary self-reported variable within the ELSA dataset. Furthermore, participants are not required to disclose further clinical information, nor are they required to complete PD symptomology assessments (e.g., the Unified Parkinson’s Disease Rating Scale [87]). Consequently, it is not possible to subtype the PD patients within the ELSA dataset, hence rendering it impossible to determine the influence of PD subtype on the relation between hearing loss and incident PD. Further studies that collect sufficient clinical data to allow for subtyping should be conducted.

It is important to also consider the potential role of addressed versus unaddressed hearing loss (i.e., the use of hearing aids) on symptom onset and the results obtained here. When accounting for the relation between hearing loss and dementia, it is increasingly acknowledged that prolonged auditory deprivation, due to hearing loss, may give rise to chronic cortical reorganisation, which hinders cognitive processes in favour of auditory perception [3,88]. Although widely debated, due to the potential influence of sociodemographic factors such as educational level and wealth [89], some evidence also suggests that correction of hearing loss, through hearing aids, may be associated with better cognition and a reduction in cognitive change [90,91]. With respect to the relation between hearing loss and PD, it may be that prolonged auditory deprivation, due to hearing loss, gives rise to chronic cortical reorganisation that favours auditory perception to the detriment of motor control, and such an effect may be in some way mitigated by correction of hearing loss (e.g., the use of hearing aids or cochlear implants). Hence, it may be important for the use of hearing aids to be controlled within analyses of the relation between hearing loss and PD. However, the nature of the dataset analysed here renders it impossible for the potential influence of addressed versus unaddressed hearing loss on symptom onset to be accounted for within this analysis. Specifically, participants are only asked to disclose whether they use a hearing aid in wave 7 (whilst completing the HearCheck Screening test). Consequently, hearing aid use cannot be added as a longitudinal covariate to the joint longitudinal and survival analysis. Therefore, these findings should be treated with a degree of caution, and further studies that control for the use of hearing aids should be conducted.

This study is not without limitations. First, incident PD cases within the final analytical sample were low (*n* = 151). However, it is unlikely that this substantially biased our findings. Specifically, in 2018, the incidence rate of PD in the English population was 0.22 percent [92]. Within this study, the incidence rate of PD was 1.05 percent. Thus, although the raw number of incident PD cases was low, this is reflective of the overall incidence rate of PD in the general population. Second, caution must be applied when generalising from the results of this study. The ELSA is a volunteer cohort [56]. Although the sample aims to be representative, there is a greater proportion of females, people from managerial/technical and skilled manual social classes, and ethnic majority groups [56,93]. Therefore, further studies using alternative, perhaps more representative, samples are required. Finally, due to the lack of repeat data regarding the use of hearing aids, it is impossible for the potential influence of hearing loss treatment (i.e., hearing aids or cochlear implants) on symptom onset to be accounted for. Therefore, further studies that collect additional data regarding hearing loss treatment are required.

## 5. Conclusions

To conclude, the present study observed that self-reported hearing loss is not a substantial risk factor for incident PD. However, sample limitations necessitate further analyses in alternative populations to substantiate this finding. Should findings be replicated, treatment of hearing loss may not substantially reduce the risk of developing PD. Additionally, exploratory analyses indicated that more objectively derived hearing loss is also not a risk factor for incident PD. However, given the small number of incident PD cases in this sample, further large-scale, planned, analyses are needed to corroborate this conclusion.

## Figures and Tables

**Table 1 brainsci-13-01196-t001:** Joint longitudinal and survival model.

	Mean	SD	Confidence Interval	*p*
Longitudinal Outcome				
Intercept	1.357	0.113	1.135–1.581	<0.001
Cohort Y => 2	1.812	0.019	1.774–1.847	<0.001
Cohort Y => 3	4.179	0.026	4.126–4.228	<0.001
Cohort Y => 4	6.338	0.040	6.263–6.417	<0.001
Age	−0.058	0.002	−0.062–−0.055	<0.001
Wave Number	−0.058	0.002	−0.062–−0.053	<0.001
Survival Outcome				
Age	0.54	0.013	0.026–0.078	<0.001
Hearing Loss	0.063	0.075	−0.080–−0.219	0.402

**Table 2 brainsci-13-01196-t002:** Baseline demographic characteristics of study cohort by hearing loss status.

		Hearing Loss Status		
	**Normal (*n* = 3575)**	**Moderate (*n* = 850)**	**Moderately Severe/Severe (*n* = 387)**	***p*** *
Sex, Male	1416	417	232	F(2,4809) = 459.9, *p* < 0.001
Age, mean (SE), years	66.48 (0.283)	72.62 (0.396)	75.95 (0.454)	F(1,4809) = 38.01, *p* < 0.001
Incident PD (*n*)	15	5	5	F(2,4809)= 2.937, *p* = 0.053

* Note. The *p*-values documented here were obtained from between-subject ANOVAs which examined whether demographic (age and sex) and PD incidence significantly differ across hearing loss groups. In this analysis, hearing loss level formed the independent variable and the relevant demographic variable formed the dependent variable.

## Data Availability

The data analysed in this study are openly available in the UK Data Service at https://doi.org/10.5255/UKDA-SN-5050-25, accessed on 11 July 2023.

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
