# Peer review of "Is Hearing Loss a Risk Factor for Idiopathic Parkinson’s Disease? An English Longitudinal Study of Ageing Analysis"

_brainsci, 2023, doi:10.3390/brainsci13081196_

Round 1
Reviewer 1 Report
Authors aim to address the question of whether hearing loss is predictive of Parkinson's Disease in a large, prospective clinical cohort. Authors provide adequate justification to explore hearing loss and the association with PD based on previous work outlining potential common mechanisms, like oxidative stress.
Major concerns are with the presentation of the data. There is little clinical interpretation of the longitudinal outcomes listed in Table 1. No statistical information is provided in Table 2 to interpret the P-value. An no where are the results of the HearCheck screen provided to internalize the results nor was it discussed the stability of hearing changes across waves. The data was collected as part of a longitudinal design but appears to be collected as a cross-sectional study.
Additionally, no where is it discussed the difference in unaddressed hearing loss versus aided hearing loss in the onset of symptoms. Auditory deprivation can have downstream consequences just as much as the underlying cause of hearing loss in relation to cognitive decline.
Reviewer 2 Report
Authors elaborate on the hearing loss and its associations with Parkinson's Disease (PD). I have a few suggestions regarding changes in the manuscript:
1. While authors elaborate on the patomechanisms of hearing loss, less is presented on the patomechanism of PD. Authors should acknowledge the possible hypotheses e.g. the neuroinflammatory
Ref.
[A] Glial reactions in Parkinson's disease. Mov Disord. 2008 Mar 15;23(4):474-83. doi: 10.1002/mds.21751. PMID: 18044695.
[B] Platelet-to-lymphocyte ratio and neutrophil-tolymphocyte ratio may reflect differences in PD and MSA-P neuroinflammation patterns. Neurol Neurochir Pol. 2022;56(2):148-155. doi: 10.5603/PJNNS.a2022.0014. Epub 2022 Feb 4. PMID: 35118638.
2. Essential tremor is a risk factor of PD, moreover it is associated with hearing loss. Authors could elaborate on this context.
3. It would be valuable to discuss possible associations of hearing loss with other parkinsonisms e.g. atypical parkinsonisms
4. The limitations of the study should be indicated in a separate paragraph.
5. Authors should elaborate whether the subtype of PD was a possibly significant feature in the vulnerability to hearing loss
Round 2
Reviewer 1 Report
Authors have addressed the areas of concerns adequately.
Reviewer 2 Report
I do not have further comments.